# Origins and Genetic Characteristics of Egyptian Peach

**DOI:** 10.3390/ijms25158497

**Published:** 2024-08-03

**Authors:** Mohamed Ezzat, Weihan Zhang, Mohamed Amar, Elsayed Nishawy, Lei Zhao, Mohammad Belal, Yuepeng Han, Liao Liao

**Affiliations:** 1State Key Laboratory of Plant Diversity and Specialty Crops, Wuhan Botanical Garden of Chinese Academy of Sciences, Wuhan 430074, China; dr_mohamed201110@mails.ucas.ac.cn (M.E.); whzhang@wbgcas.cn (W.Z.); mohamed.amar@wbgcas.cn (M.A.); elnishawy@mail.hzau.edu.cn (E.N.); zhaolei@wbgcas.cn (L.Z.); maabdrc@mails.ucas.ac.cn (M.B.); 2Plant Genome Laboratory, Department of Genetic Resources, Desert Research Center, El-Matareya, Cairo 11753, Egypt; 3University of Chinese Academy of Sciences, Beijing 100049, China

**Keywords:** peach, origin, Egypt, SNP, IBD, population structure

## Abstract

Peach (*Prunus persica*), a significant economic fruit tree in the Rosaceae family, is extensively cultivated in temperate and subtropical regions due to its abundant genetic diversity, robust adaptability, and high nutritional value. Originating from China over 4000 years ago, peaches were introduced to Persia through the Silk Road during the Han Dynasty and gradually spread to India, Greece, Rome, Egypt, Europe, and America. Currently grown in more than 80 countries worldwide, the expansion of peach cultivation in Egypt is mainly due to the development and utilization of peach varieties with low chilling requirements. These varieties exhibit unique phenotypic characteristics such as early maturity, reduced need for winter cold temperatures, low water requirements, and high economic value. In this study, a systematic analysis was conducted on the genetic characteristics and kinship relationships of peaches with low chilling requirements in Egypt. We conducted a comprehensive evolutionary and Identity-by-Descent (IBD) analysis on over 300 peach core germplasm resources, including Egyptian cultivars with low chilling requirements, to investigate their origin and genetic characteristics. The evolutionary analysis revealed that ‘Bitter almond’ is closely related to China’s wild relative species *Prunus tangutica* Batal, while ‘Early grand’ shares one branch with Chinese ornamental peach cultivars, and ‘Nemaguard’ clusters with some ancient local varieties from China. The IBD analysis also indicated similar genetic backgrounds, suggesting a plausible origin from China. Similarly, the analysis suggested that ‘Swelling’ may have originated from the Czech Republic while ‘Met ghamr’ has connections to South Africa. ‘Desert red’, ‘Early swelling’, and ‘Florida prince’ are likely derived from Brazil. These findings provide valuable insights into the genetic characteristics of Egyptian peach cultivars. They offer a significant foundation for investigating the origin and spread of cultivated peaches worldwide and serve as a valuable genetic resource for breeding low chilling requirement cultivars, which is of considerable significance for the advancement of peach cultivation in Egypt.

## 1. Introduction

*Prunus persica* (L.) Batsch, commonly known as peach, is a popular stone fruit belonging to the Rosaceae family [1]. The Rosaceae family is extensively distributed across subtropical regions and warm temperate areas, encompassing over 80% of deciduous fruit species [2,3,4]. Peach is the third most economically important temperate fruit worldwide, with an annual production of approximately 25 million tons (FAOSTAT, 2022). Peach cultivation is mainly concentrated in temperate regions between latitudes 30 and 45° N and S, where sufficient winter chilling is essential for successful flowering and fruit set [5]. The origin of peach can be traced back to southwestern China, where it has been domesticated for over 4000 years, as supported by genetic and phenotypic data [6,7]. China has the largest peach landraces and wild relatives, significantly influencing international peach breeding programs [8]. This abundant genetic diversity provides valuable genetic resources for enhancing disease and pest resistance, improving fruit size and quality, and prolonging postharvest shelf life [9]. The domesticated peach from China was spread westward through the ancient Silk Road, passing through Persia (modern-day Iran) and eventually reaching Europe [10]. Regions such as the Mediterranean, including Egypt, likely introduced peaches from Persia during that period [10,11].

Egypt ranks eighth in global peach production and first in Africa [12], emerging as a potential exporter. This rise can be attributed to varieties that meet the low chilling requirement (CR), exhibit early maturity, high yield, and offer exceptional fruit quality, which drives increased demand from Europe and Arab Gulf countries [13,14]. The chilling requirement (CR) refers to the minimum hours of low temperatures needed for dormancy release and growth initiation in spring. Global warming has led to elevated winter temperatures, resulting in fewer chilling hours and abnormal blossoming. These changes significantly threaten peach cultivation, impacting dormancy release, fruit yield, and quality [15]. Under warmer winter conditions, low chilling requirement cultivars may exhibit consistent productivity, while high chilling requirement cultivars may face challenges in blooming and fruit set [16]. Developing high-quality peach varieties with low chilling requirements has recently enabled large-scale cultivation in countries like Egypt, which have limited low temperature accumulation. 

Egypt is well known for its botanical diversity and valuable genetic resources, positioning it among the top Arab nations in agricultural production [14]. The peach cultivation industry in Egypt has grown rapidly, especially in regions such as the Alexandria, Al-Qalyubia, Upper Egypt, and North Sinai Governorates [17]. This growth is attributed to the availability of low chilling requirement cultivars that allow early harvest, making them highly suitable for export [18]. Egyptian peaches have advantages in early maturation, high productivity, superior fruit quality, and competitive export pricing. As a result, Egypt is emerging as a potential contender in the global peach market. Egypt boasts diverse peach varieties from different origins, each with unique characteristics. The local cultivar ‘Met ghamr’ is known for its adaptability to various environmental conditions [19]. Meanwhile, international peach varieties offer varying quality traits and maturity periods, catering to specific environmental and market needs [20]. Additionally, rootstocks like ‘Bitter almond’ are commonly used in limestone lands with rainfall irrigation systems. The global rootstock ‘Nemaguard’ provides nematode resistance and compatibility with grafted peach varieties [21].

Studying fruit tree genetics facilitates a more comprehensive understanding of phylogenetic relationships, genetic variation, and the domestication origin of populations [22]. Whole-genome re-sequencing has opened up new opportunities and strategies for research in related fields, accelerating our insights into the origin, domestication, and global spread of specific fruit trees [23]. Despite the economic and nutritional importance of Egyptian peaches, limited studies have been conducted to understand their genetics compared to Chinese peach varieties. Previous studies have primarily focused on agronomic traits and phenotypic variations, leaving a significant gap in understanding crucial genetic insights, such as nucleotide variations, structural variants, and evolutionary relationships within Egyptian peach cultivars. Limited genetic studies utilizing SSR, ISSR, and SCoT markers have highlighted the genetic diversity of these cultivars [24,25]. Sayed et al. [26] identified genetic diversity and phylogenetic relationships within the Prunus genus using cpDNA barcoding, while Nabil et al. [27] evaluated the ‘Early Swelling’ peach cultivar, revealing genetic variability through ISSR and RAPD analyses. Our research objectives include: 1. Conducting a comprehensive genomic analysis of nucleotide variants in eight Egyptian peach cultivars. 2. Investigating the genetic diversity and structure within these cultivars. 3. Assessing the evolutionary relationships and potential historical diffusion patterns. This study aims to enhance the understanding of the genetic foundation of low chilling Egyptian peaches, thus contributing valuable information for future breeding programs and cultivation strategies. 

## 2. Results

### 2.1. Geographical Distribution and Morphological Characteristics of Egyptian Peaches

Egypt is located within the subtropical climatic zone, spanning between latitudes 22° and 32° N and longitudes 25° and 35° E. This location provides favorable environmental conditions for peach cultivation, particularly in the Alexandria, Al-Qalyubia, and North Sinai Governorates. Among these regions, Alexandria, in particular, stands out as a suitable location. Over the past two decades, Egyptian peach cultivation has predominantly relied on low chilling foreign cultivars like ‘Florida prince’, ‘Early swelling’, ‘Desert red’, and ‘Swelling’. These cultivars have successfully adapted to local conditions, showing early, mid-, or late ripening periods and desirable fruit characteristics. In this study, eight Egyptian peach cultivars were collected from various locations across Egypt. Precise geographic coordinates were recorded for each collection site to ensure accurate spatial representation of the sampling points shown in Figure 1 and Table 1.

‘Florida prince’ is the predominant peach cultivar in Egypt due to its early ripening, excellent firmness, and vibrant coloration. It exhibits adaptability to local environmental conditions, allowing it to appear in markets ahead of other fruits by fulfilling its low chilling requirement for early flowering and fruit maturation. This cultivar has red skin, yellow flesh, medium-sized fruits, a sticky core, high flesh hardness, and it reaches maturity during the first week of April after receiving 100–150 h of cold winter chilling. ‘Early swelling’ features red-whitish skin, white flesh, loose core, low flesh firmness, medium fruit size, ripening in the third week of April, and requiring 150–250 chilling hours. ‘Desert red’ is characterized by its red skin, yellow flesh, medium flesh hardness, large fruit size, and ripening in mid-May with 200–250 chilling hours. ‘Swelling’ has red-whitish skin, white flesh, loose core, high sugar content, and ripens in early June, requiring 200–250 chilling hours. Lastly, ‘Early grand’, cultivated in El-Sheikh Zuweid at North Sanai, has reddish-yellow skin and yellow flesh. It boasts large fruit size and high flesh hardness, ripening in late April with a chilling requirement of 150–250 h. Despite the dominance of foreign cultivars, some seedy plantations of the local ‘Met ghamr’ still exist in Al-Qalyubia. This variety was once Egypt’s primary peach cultivar due to its exceptional adaptation to local conditions, distinctive taste, and aroma, appearing in markets late in the season.

### 2.2. Sequencing and Mapping of Egyptian Peach Cultivars

DNA sequencing was performed on various Egyptian peach cultivars, including the local cultivar ‘Met ghamr’ and international cultivars ‘Florida prince’, ‘Early swelling’, ‘Early grand’, ‘Desert red’, and ‘Swelling’, as well as the global rootstock ‘Nemaguard’, and a local ‘Bitter almond’ rootstock. Following preprocessing to remove low-quality bases and adapters, a total of 171.00 Mb clean reads were generated, corresponding to 50.81 Gbp clean data. The average Q30 value was 93.4%, indicating sequencing accuracy and high-quality data. The average GC content of the retained reads was 39% and about 94% of the clean reads mapped successfully to the peach reference genome (v.2.1). Each cultivar was sequenced to a depth of 1× resulting in a genome coverage of 95%. These results demonstrate the reliability of the sequencing data (Table 2).

### 2.3. Genome-Wide Nucleotide Variants

Alignment of clean reads from eight Egyptian peach cultivars to the peach reference genome revealed various nucleotide variants, predominantly single-nucleotide polymorphisms (SNPs). Notably, ‘Bitter almond’ showed approximately four-fold more SNPs than other cultivars. SNPs were categorized into transitions (Ts) and transversions (Tv) based on the types of nucleotide substitution, with Ts accounting for 65.3% and Tv for 34.7%. Heterozygous SNPs were more common (52.31%) than homozygous SNPs (47.69%) (Figure 2 and Appendix A). The Ts/Tv ratio varied among cultivars, with ‘Met ghamr’ having the highest ratio at 1.97 and ‘Florida prince’ having the lowest at 1.82. The average Ts/Tv ratio across all peach cultivars was 1.88. In addition to SNPs, structural variants (SVs) like insertions (INS), deletions (DEL), inversions (INV), intra-chromosomal translocations (ITX), and inter-chromosomal translocations (CTX) were also identified. Deletions were the most prevalent type of SV, accounting for 48%, followed by insertions with 26% (Figure 3). The count of SVs varied across different peach cultivars: ‘Met ghamr’ (9971 SVs), ‘Early grand’ (9747 SVs), ‘Early swelling’ (10,549 SVs), ‘Florida prince’ (10,939 SVs), ‘Swelling’ (10,892 SVs), ‘Desert red’ (10,752 SVs), ‘Nemaguard’ (9405 SVs), and ‘Bitter almond’ (22,687 SVs) (Appendix A). The total SNPs and SVs observed within the studied peach cultivars were 8,701,042 and 94,942, respectively. 

### 2.4. Annotation of Nucleotide Variants

During annotation, SNP distribution within the peach genome was examined and analyzed (Appendix A). SNPs were found primarily in intergenic regions (48%) and within 5 Kb downstream regions (39%), indicating a predominant presence in intergenic and downstream regions (Figure 4A). SNPs within coding region (CDS) were classified as synonymous or nonsynonymous variants. Nevertheless, these subtle genetic changes can exert significant impacts on organismal traits. An average of 21,764 synonymous SNPs and 25,364 nonsynonymous SNPs were annotated with a ratio of 1.16, higher than in *Arabidopsis thaliana* (0.83). Similar to SNP annotation, InDels were also mainly distributed in intergenic (43%) and downstream regions (45%) (Appendix A). In the coding region (CDS), there were 1332 frameshift mutations, 254 codon deletions, and 221 codon insertions (Figure 4B). Variants in the coding regions, particularly nonsynonymous SNPs and frameshift InDels, can influence protein structure and function, highlighting their importance. The findings presented herein elucidate the distribution and functional impact of SNPs and InDels in peach genetic variation and evolution.

### 2.5. Phylogenetic Analysis and Origin of Egyptian Peach Cultivars

To elucidate the evolutionary status and genetic relationships between Egyptian peaches and other core peach germplasm resources, we retrieved 301 DNA re-sequencing datasets of core peach materials from NCBI, including 123 wild and closely related species, 32 ornamental peaches, 24 local varieties, and 122 modern cultivated peach. These datasets were combined with the previously sequenced data of eight Egyptian peach cultivars (Appendix A). Subsequently, their genetic relationships were analyzed through phylogenetic analysis using SNPs. The genetic relationships and backgrounds of individuals were investigated by constructing a phylogenetic tree using the neighbor-joining method, which revealed three major clusters, each containing distinct sub-clusters based on genetic backgrounds and geographic origins (Figure 5). Cluster I included wild relatives, divided into three sub-clusters: *P. tangutica* Batal accessions, *P. mira* Koehne genetic background, and *P. kansuensis* Rehder genetic background. Cluster II was dominated by ornamental accessions from eastern-central and northern China, with a separate sub-cluster comprised of landraces from southwestern China. Cluster III encompassed improved accessions from diverse regions, with one sub-cluster consisting of accessions from America, Europe, and South Africa, and another comprising accessions primarily from China and Japan. 

Egyptian peach cultivars exhibited diverse genetic relationships within the Prunus species, with each cultivar demonstrating unique clustering patterns and associations with specific groups. ‘Bitter almond’ clustered with the wild *P. tangutica* from China, ‘Early grand’ with eastern-central Chinese ornamentals, and ‘Nemaguard’ with the southwestern Chinese landraces. ‘Swelling’ grouped with the Czech Republic’s Golo genotype, and ‘Met ghamr’ demonstrates close genetic relationships with improved accessions from South Africa (Everts, Reigels, C17, and C18) and Canada (Harrow Blood-1 and Harrow Blood-2). ‘Desert red’, ‘Early swelling’, and ‘Florida prince’ were part of the improved group, clustering with Brazilian accessions (Coral, Chirva, and Charme). The origins of Egyptian peaches reveal intriguing patterns and diverse geographic sources, suggesting a historical connection between Egypt and China for ‘Bitter almond’, ‘Early grand’, and ‘Nemaguard’. Additionally, ‘Swelling’ exhibits origins connected to the Czech Republic, and ‘Met ghamr’ is linked to South Africa and Canada. Meanwhile, the origins of ‘Desert red’, ‘Early swelling’, and ‘Florida prince’ are traced back to Brazil. These results provide a basis for understanding the origin and genetic characteristics of the Egyptian peach cultivars, reveal their evolutionary status, and provide insights into the domestication history and genetic diversity of the peach.

### 2.6. Population Structure

The genetic composition of the peach resource population was examined through population structure analysis. At K = 2, the population can be classified into wild species and cultivated peaches. Additionally, when K ranges from 4 to 7, wild species can be distinctly categorized into *P. mira* Koehne, *P. davidiana* (Carrière) Franch, and *P. kansuensis* Rehder, indicating significant genetic variations between these three types of wild peaches. Moreover, at K = 5, genetic differences between local and improved cultivated peaches were observed. At K = 7, there was a clear differentiation between local ornamental and edible peach varieties (Figure 6A). The analysis revealed the genetic backgrounds of the peach resources, aiding in understanding their origin and domestication.

### 2.7. Principal Component Analysis (PCA)

Principal component analysis (PCA) was conducted on the variation dataset using TASSEL software v.3.0 to explore peach population structure. PCA revealed that PC1 and PC2 exhibited significant differentiation, accounting for a cumulative proportional variance of 79.4%. The PCA analysis supported the results obtained from the phylogenetic tree (Figure 5) and confirmed the stability of the population structure analyses (Figure 6A). In the PCA plot, the accessions formed two relatively clustered groups, with a total molecular variation explained by these groups at 49.6% and 29.8%, respectively (Figure 6B). Cluster I represented wild relatives, including the Egyptian ‘Bitter almond’, closely clustered within this group. Cluster II included ornamental, landrace, and improved accessions, with the remaining seven Egyptian cultivars showing a closer relationship within a distinct group near the PC2 axis. The consistency between PCA and structure analyses reinforced the understanding of the peach population structure and evolutionary history. These comprehensive analyses, including origin studies, phylogenetic assessments, and population structure analyses using PCA and structure methods, provided comprehensive insights into the origins and genetic diversity of the low chilling requirement peaches cultivated in Egypt within the Prunus species. The findings illuminate the relationships between the different peach varieties and can help protect valuable germplasm resources.

### 2.8. Identity-by-Descent (IBD) Analysis

The identity-by-descent (IBD) relationship patterns among the 309 peach accessions were analyzed (Figure 7 and Appendix A). IBD values from pairwise peach comparisons were produced. ‘Bitter almond’, ‘Early grand’, and ‘Nemaguard’, showed high IBD values with Chinese varieties, namely, ‘Da ba dan’, ‘Tanchun’, and ‘Jiuyang qing tao’, with respective values of 0.72, 0.67, and 0.63, suggesting an origin from China. ‘Swelling’ had the highest IBD value of 0.66 with the Czech peach ‘golo’, suggesting a potential origin from the Czech Republic. ‘Met ghamr’ showed the highest IBD value of 0.70 with South African peach ‘Everts’, suggesting a South African origin. Notably, ‘Desert red’, ‘Early swelling’, and ‘Florida prince’ formed a distinct cluster with high IBD values with Brazilian peaches, indicating their likely origin in Brazil. These findings provide valuable insights into the geographical origins and genetic lineages of Egyptian peach cultivars, reflecting a rich history of international exchange and breeding practices in Egyptian peach cultivation.

## 3. Discussion

This study analyzed nucleotide variants in eight Egyptian peach cultivars by aligning clean reads to the peach reference genome. Our findings revealed a wide range of nucleotide variations, including single-nucleotide polymorphisms (SNPs), InDels, and structural variants (SVs). ‘Bitter almond’ showed significantly higher SNPs than other Egyptian peach cultivars, indicating greater genetic diversity within this variety. Most SNPs were biallelic, with transitions (Ts) accounting for 65.3% and transversions (Tv) for 34.7%. The Ts/Tv ratio of 1.88 in our study is higher than in other plant species, such as 1.46 in grapes [28], 1.5 in potatoes [29], and 1.27 in apples [30]. This divergence may suggest unique evolutionary and genetic dynamics in peach cultivars. Keller et al. [31] proposed that a bias towards transitions (Ts) may be attributed to stronger purifying selection against transversions (Tv), a phenomenon that can vary across different organisms [32].

Heterozygous SNPs (52.31%) were more frequent than homozygous SNPs (47.69%), indicating the presence of allelic variation and higher genetic diversity within the studied peach cultivars. The distribution of SNPs in intergenic and downstream regions suggests their potential involvement in gene expression regulation and biological traits [33]. Additionally, identifying nonsynonymous SNPs, which cause amino acid changes, implies functional significance and phenotypic diversity among the peach cultivars [34]. Moreover, we investigated the frequency of InDels in the coding sequence (CDS), as they can potentially disrupt gene function. Interestingly, we observed that InDels in CDS were less frequent than nonsynonymous SNPs. ‘Bitter almond’ had four-fold more frameshift mutation than other cultivated peaches, potentially contributing to phenotypic variation observed among prunus varieties [35]. Our investigation into structural variants (SVs) such as deletions, insertions, and inversions within or between chromosomes revealed a higher prevalence in ‘Bitter almond’. These SVs play a significant role in shaping the genomic landscape and contributing to phenotypic diversity [36,37,38], suggesting potential regulatory or functional impacts on gene expression [39].

The evolutionary relationships and diversification patterns within the prunus species have been the subject of previous studies [40,41]. This study conducted phylogenetic tree and population structure analyses to gain insights into the genetic diversity, relationships, and evolutionary history of Egyptian peach cultivars within the prunus species. These findings are significant for understanding the domestication and origin of low chilling Egyptian peach cultivars. By classifying different peach groups and clustering cultivars with wild relatives, ornamental varieties, landraces, and improved cultivars, we can better highlight their unique genetic backgrounds, origins, and evolutionary history. These results are consistent with previous study [7], which also supports the presence of distinct genetic groups within peach populations. The clustering of the Egyptian ‘Bitter almond’ with wild relatives from China suggests a closer relationship to wild peach species. This indicates that ‘Bitter almond’ has retained characteristics of wild ancestors and has not undergone extensive domestication, showing a phylogenetic distance between almond and peach species. ‘Early grand’ clustering with the ornamental group suggests a shared genetic background and potential common ancestry with ornamental peach varieties. ‘Nemaguard’ clustering with landraces indicates a closer genetic relationship to locally adapted peach varieties in China, implying specific environmental adaptability or unique characteristics beneficial for its role as a rootstock in peach cultivation [42]. ‘Nemaguard’ is believed to be a seedling selected from seeds obtained in 1949 from a commercial importer, *Prunus davidiana*. Although its tree and fruit characteristics closely resemble those of peach (*P. persica*), it is believed to have a hybrid origin, potentially resulting from a cross between *P. persica* and *P. davidiana* [43]. The IBD analysis for ‘Bitter almond’, ‘Early grand’, and ‘Nemaguard’ also indicated similar genetic backgrounds, suggesting a plausible origin from China.

According to the phylogenetic tree and IBD analysis, the ‘Met ghamr’ variety shows the highest IBD values compared to the South African peach varieties, indicating a potential origin from South Africa. ‘Desert red’, ‘Early swelling’, and ‘Florida prince’ form a distinct cluster with the highest IBD values with Brazilian peaches, suggesting their probable origin in Brazil. Additionally, ‘Swelling’ shows the highest IBD values compared to the Czech peach variety, hinting at a potential origin from the Czech Republic. These findings reveal international exchanges and introductions of peach accessions into Egypt, providing opportunities for further exploration of trade routes and mechanisms in the spread of peach varieties. The peach has undergone selection and adaptation, resulting in locally adapted accessions widely distributed across Europe, Asia, South Africa, Australia, and the Americas. These historical movements and exchanges of peach varieties have enriched genetic diversity, facilitating the development of low-chill varieties adapted to various regions [10,44]. 

Tracing the origins of peaches in Egypt can be linked back to their domestication in China, where wild peach trees native to Tibet and western China were first domesticated around 4000 years ago [45]. From China, peaches spread along ancient trade routes to Persia (modern-day Iran), where they gained prominence [46]. The peach’s migration from Persia to Greece during classical times, around 300–400 BCE [47], likely facilitated the introduction of peaches to the Mediterranean region, including Egypt [10,11]. During the Greco-Roman period, peaches were introduced to the Mediterranean basin [10], where the Romans played a crucial role in the dissemination of peaches throughout the empire. Following their occupation of the western coast of Persia (modern-day Syria) [48], they transported peaches across various Roman provinces, including Egypt [49]. Agricultural practices and crop diffusion during the Islamic Golden Age further contributed to the establishment of peach orchards in North Africa, including Egypt, Morocco, and Algeria [50]. This historical trajectory illustrates how peaches, originally domesticated in China, eventually became an essential fruit in Egypt through a complex network of trade, expansions, and agricultural advancements. 

Studying the genetic diversity among peach accessions is crucial for breeders to select and enhance desirable traits and preserve genetic resources. Breeders can develop novel varieties with improved characteristics by harnessing the distinctive genetic diversity within Egyptian peach cultivars and other peach accessions. Given Egypt’s unique geographical location and climatic conditions, only peach cultivars with low chilling requirements are suitable for fostering Egypt’s peach industry. With the increasing impacts of climate change, Egypt’s expertise in cultivating low chill requirement varieties positions it well to support and expand these cultivars to other Mediterranean countries. As temperatures rise and traditional peach-growing regions face greater challenges in meeting chilling requirements, low chill varieties could become crucial for sustaining peach production.

## 4. Materials and Methods

### 4.1. Sample Collection of Egyptian Peach Cultivars

The present study employed eight Egyptian peach cultivars collected from different locations in Egypt. Young leaves were sampled during the spring and cryopreserved in liquid nitrogen for subsequent DNA extraction and sequencing. The georeferencing of samples was conducted using the Map Factor GPS Navigator Maps (MapFactor GPS Navigation App), as shown in Table 1. The Egyptian peach samples selected for re-sequencing encompassed the following: ‘Met ghamr’ as the local cultivar, ‘Florida prince’, ‘Early swelling’, ‘Early grand’, ‘Desert red’, and ‘Swelling’ as foreign cultivars, ‘Nemaguard’ as a global rootstock, and ‘Bitter almond’ as a local rootstock. In addition to these sequenced samples, we downloaded 301 peach core germplasm materials from NCBI for use in the subsequent relevant analysis. Detailed sample information is shown in Appendix A.

### 4.2. Extraction and Purification of Genomic DNA

DNA extraction from the Egyptian peach cultivars was conducted using the DNeasy Mini Kit (Qiagen Santa Clarita, CA, USA). The extraction procedure followed the manufacturer’s instructions, which included grinding the tissue sample under liquid nitrogen. Subsequently, 400 μL of buffer AP1 and 2 μL of RNase A stock solution (100 mg/mL) were added to the ground tissue and vortexed vigorously. The mixture was then incubated at 65 ºC for 30 min with intermittent mixing. Following this, buffer AP2 (130 μL) was added to the lysate, mixed, and incubated on ice for 5 min. The lysate was applied to the QIAshredder mini spin column and centrifuged at 10,000 rpm for 2 min. Next, a 1.5 volume of buffer AP3/E was added to the cleared lysate and mixed by pipetting. A volume of 650 μL of the mixture, including any precipitate that may have formed, was applied to the DNeasy mini spin column sitting in a 2 mL collection tube. The mixture was then centrifuged at 8000 rpm for 1 min, and flow-through was discarded. A 500 μL volume of buffer AW was added to the DNeasy mini spin column and centrifuged at 8000 rpm for 1 min. Buffer AW 500 μL was added to the DNeasy mini spin column and centrifuged at 10,000 rpm for 2 min to dry the membrane. The DNeasy mini spin column was transferred to a 1.5 mL microcentrifuge tube, and 100 μL of buffer AE was pipetted directly onto the DNeasy membrane. The microcentrifuge was incubated for 5 min at room temperature and then centrifuged at 10,000 rpm for 1 min to elute. The quality and integrity of the DNA were examined using the NanoDrop™ ND-1000 (Thermo Fisher Scientific, Waltham, MA, USA), followed by electrophoresis in 0.8% agarose gel. High-molecular-weight DNA aliquots with 230/260 and 260/280 ratios ranging between 1.8 and 2.0 and 1.8 and 2.2 were then sent to BMKGENE Co. (Beijing, China) for library construction and sequencing.

### 4.3. Library Construction and Sequencing

The libraries had an insert size of 350 bp, and the pair-end reads were 150 bp in length. All libraries were sequenced using the Illumina HiSeq 2500 platform (Illumina, San Diego, CA, USA). The raw reads underwent trimming for the removal of low-quality sequences and adaptors using Trimmomatic v.0.36 [51]. Subsequently, clean reads obtained from each peach cultivar were aligned to the *Prunus Persica* reference genome v.2.1.

### 4.4. Read Mapping and Variants Calling

The clean data were mapped to the *P. persica* reference genome using the Burrows–Wheeler Alignment (BWA) algorithm v.0.7.15 [52]. Subsequently, the mapped data underwent sorting and duplication marking using Picard (https://broadinstitute.github.io/picard, accessed on 1 April 2023). The Genome Analysis Toolkit (GATK) v.3.8 was utilized for filtering and detecting SNPs and InDels between peach accessions and the reference genome [53]. To improve the accuracy of variant-calling and address errors caused by incorrect mapping, VCFtools v.0.1.14 were used to extract the original high-quality variants [54]. Low-quality SNPs and InDels were filtered from our dataset based on exclusion criteria such as minimum minor allele frequency (MAF < 5%) and missing rate (MD > 10%). Subsequently, the data were converted into a Variant Call Format (VCF). Meanwhile, SVs, including deletions, insertions, inversions, intra-chromosomal translocations, and inter-chromosomal translocations, were identified using the BreakDancer software package v.1.4.5 [55]. 

### 4.5. Functional Annotation of Genetic Variants

Annotation of gene-based SNPs and InDels in the peach genome was conducted utilizing the ANNOVAR package (version 2013-08-23) [56]. The annotation process involved categorizing SNPs and InDels based on their genomic locations relative to the reference genome annotation, including exonic (within coding regions), splicing sites (within splice junctions), 5′ and 3′ untranslated regions (UTRs), intronic (within introns), upstream (region upstream from the transcription start site), downstream (downstream from the transcription stop site), or intergenic regions (located between genes). Within coding regions, SNPs were further classified into synonymous (sSNPs), which do not alter the amino acid sequence, and nonsynonymous (nsSNPs), which alter the amino acid sequence. InDels in the coding regions were identified by their ability to cause frame-shift mutations, insertions, or deletions. Finally, heterozygosity was calculated using VCFtools v.0.1.14 [54]. 

### 4.6. Phylogenetics, Population Structure, and Principal Component Analysis (PCA)

A total of 8,701,042 high-quality SNPs were selected for phylogenetic analysis, population structure, and principal component analysis (PCA). A neighbor-joining tree was constructed using MEGA 11 to elucidate the evolutionary relationships of Egyptian peach cultivars at a genome-wide level. Population structure was evaluated using the Structure program [57]. Admixture software v.1.3.0 was used to analyze the population structure of 309 Prunus accessions, with K values ranging from 2 to 7, to determine the optimal number of subpopulations. Principal component analysis (PCA) was used to evaluate the genetic structure of the peach populations. The SmartPCA program within the Eigensoft software 6.0 package was used for PCA analysis with default parameters [58].

### 4.7. Identification of Identity-by-Descent (IBD) Segments

The IBD detection pipeline utilized a matrix consisting of 309 individuals, where their genotypes were analyzed at 8,701,042 genomic variation sites, as mentioned above. The phasing of all subjects was conducted using the fastPhase function in Beagle version 3.3.2. The extraction of pairwise shared haplotypes was performed utilizing the Beagle fastIBD function, following the methodology described by Browning and Browning [59]. The process of phasing and detecting IBD was performed independently ten times, and the identified IBD tracts were combined from all ten runs, following the authors’ recommendation. Partially intersecting sequences were extracted, and the IBD sequences with the most favorable probability scores were included in the collection of IBD tracts. The IBD detection process was conducted through ten cycles, each utilizing distinct thresholds to determine the assignment of IBD to the haplotypes of two individuals. The observed numbers ranged from zero pairwise IBD tracts to complete IBD genomes. To assess the occurrence of shared haplotypes across various genomic regions, we partitioned the genome into 10,000 bp segments and calculated the count of identified IBD tracts between the two individuals for each segment. As the number of pairwise comparisons differed between the individuals, these numbers were normalized, ranging from 0 to 1. Relative IBD was then calculated by extracting the normalized IBD per bin for the two individuals [60]. Pairs of accessions are considered to be genetically identical if they have the highest IBD value.

## 5. Conclusions

The cultivation of peach (*Prunus persica*) holds a significant position in the global fruit industry, ranking as the third most economically important temperate fruit with an annual production of approximately 25 million tons. The success of peach cultivation is closely linked to temperate and subtropical regions, where sufficient winter chilling is essential for successful flowering and fruit development. However, the challenges posed by global warming, resulting in warmer winters and reduced chilling hours, have emerged as a threat to peach production. Low chilling requirement cultivars exhibit potential for better adaptation to warmer conditions, contrasting with high chilling requirement varieties that face obstacles in blooming and fruit development under such altered climatic conditions. Peaches have been cultivated in China for over 4000 years and have subsequently spread to diverse regions such as Persia, America, and Mediterranean countries, including Egypt. The expansion of peach cultivation in Egypt can be attributed to the adoption of peach cultivars with low chilling requirements. Research efforts focusing on enhancing the performance of these low chilling requirement cultivars are pivotal for ensuring sustainable production of temperate fruits amidst the challenges posed by climate change. This study investigated the genomic variation, phylogenetic relationships, and genetic diversity among Egyptian peach cultivars with low chilling requirements. The findings of this study provide valuable insights into the domestication and global dissemination of peaches. Through phylogenetic analysis and pedigree identification (IBD) analysis, significant genetic relationships between the studied varieties were unveiled, shedding light on the probable origin of Egyptian peaches, the characteristics of their worldwide distribution, as well as the intricate evolutionary pathways that contributed to this genetic diversity.

## Figures and Tables

**Figure 1 ijms-25-08497-f001:**
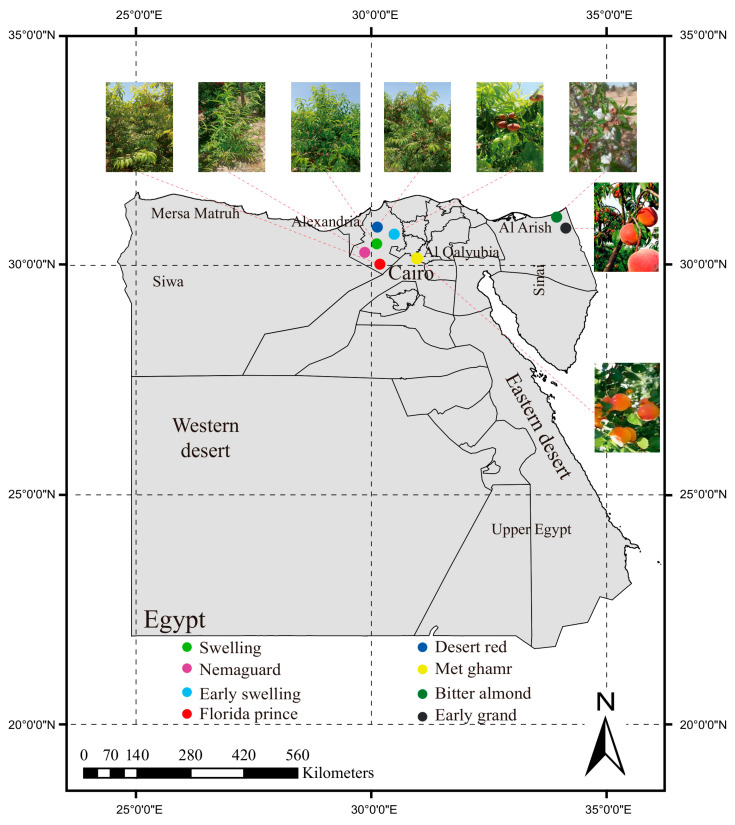
**Geographical distribution and morphological characteristics of Egyptian peach cultivars**. Map displays the spatial distribution and morphological characterization of eight Egyptian peach cultivars: Swelling (green), Desert red (dark blue), Nemaguard (purple), Met ghamr (yellow), Early swelling (blue), Bitter almond (dark green), Florida prince (red), and Early grand (black). The sampling points for these cultivars are predominantly located in the upper-latitude regions of Egypt, showing a concentrated distribution in three key regions: Alexandria, Al-Qalyubia, and North Sinai.

**Figure 2 ijms-25-08497-f002:**
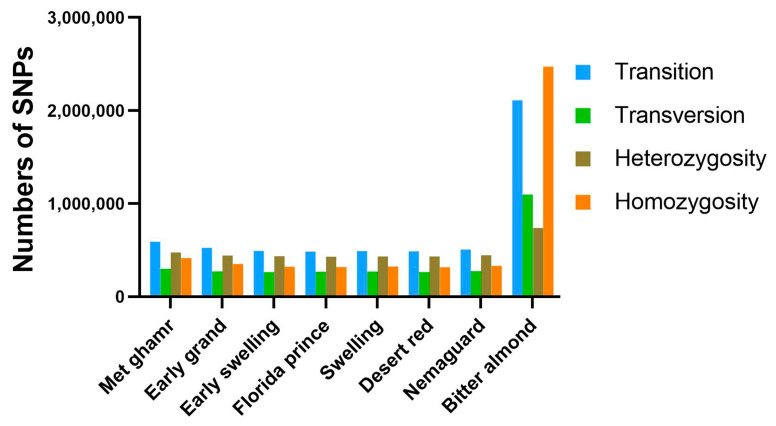
**Histogram of single-nucleotide polymorphisms (SNPs) in the Egyptian peach cultivars.** The histogram illustrates the identification and distribution of SNPs identified in the eight Egyptian peach cultivars. The SNPs are categorized into transition and transversion based on the types of nucleotide substitution. The analysis also distinguishes between heterozygous and homozygous SNPs across all peach cultivars. The colors in the histogram represent blue for transition, green for transversion, brown for heterozygous SNPs, and orange for homozygous SNPs.

**Figure 3 ijms-25-08497-f003:**
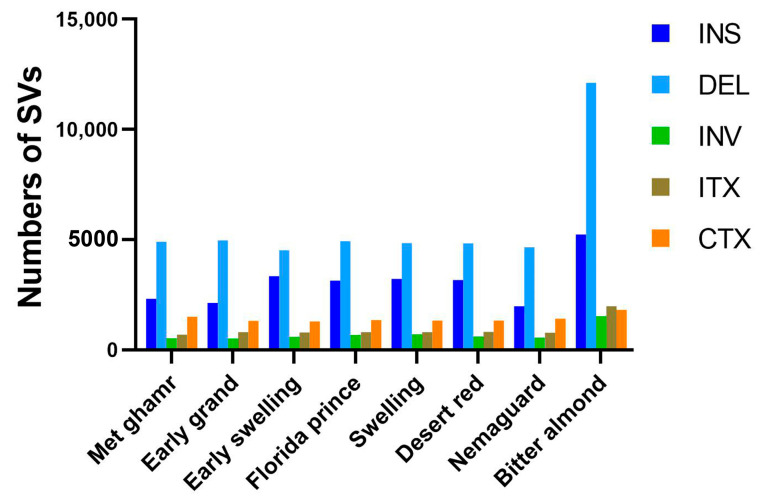
**Histogram of structural variations (SVs) in the Egyptian peach cultivars.** The histogram depicts the identification and distribution of SVs identified in the eight Egyptian peach cultivars: insertions (INS), deletions (DEL), inversions (INV), intra-chromosomal translocations (ITX), and inter-chromosomal translocations (CTX). Among these, DEL (depicted in light blue) emerged as the most prevalent type of SV, followed by INS (depicted in dark blue). The other types of SVs, INV, ITX, and CTX, are represented in green, brown, and orange, respectively.

**Figure 4 ijms-25-08497-f004:**
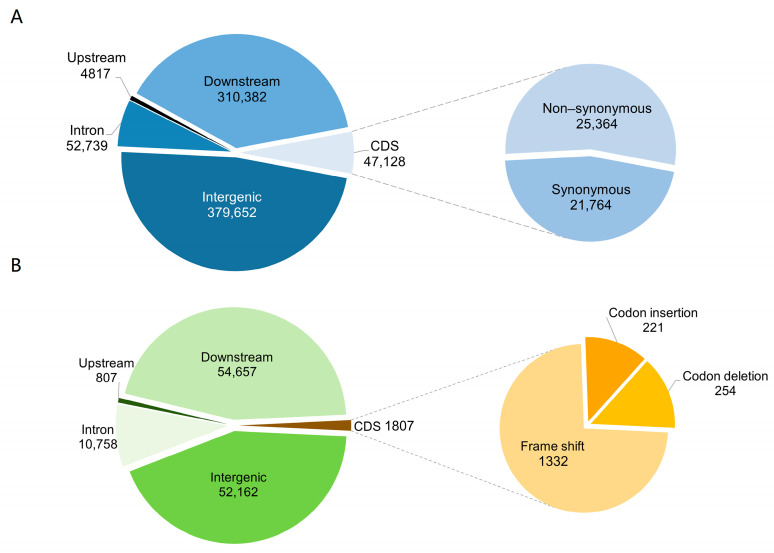
**Distribution and annotation statistics of SNPs and InDels in the peach genome**. (**A**) displays the annotation of gene-based SNPs. The SNPs are categorized based on their genomic locations relative to the reference genome annotation, including intronic regions (within introns), upstream (before the transcription start site), downstream (after the transcription stop site), and intergenic regions (between genes). Within the coding regions, SNPs are subdivided into synonymous SNPs (sSNPs), which do not alter the amino acid sequence, and nonsynonymous SNPs (nsSNPs), which alter the amino acid sequence. (**B**) illustrates the annotation of gene-based InDels. Similar to SNPs, InDels are categorized based on their genomic locations relative to the reference genome annotation. InDels are further identified in the coding regions by their potential to cause frameshift mutations (which alter the gene’s reading frame) or specific codon insertions and deletions.

**Figure 5 ijms-25-08497-f005:**
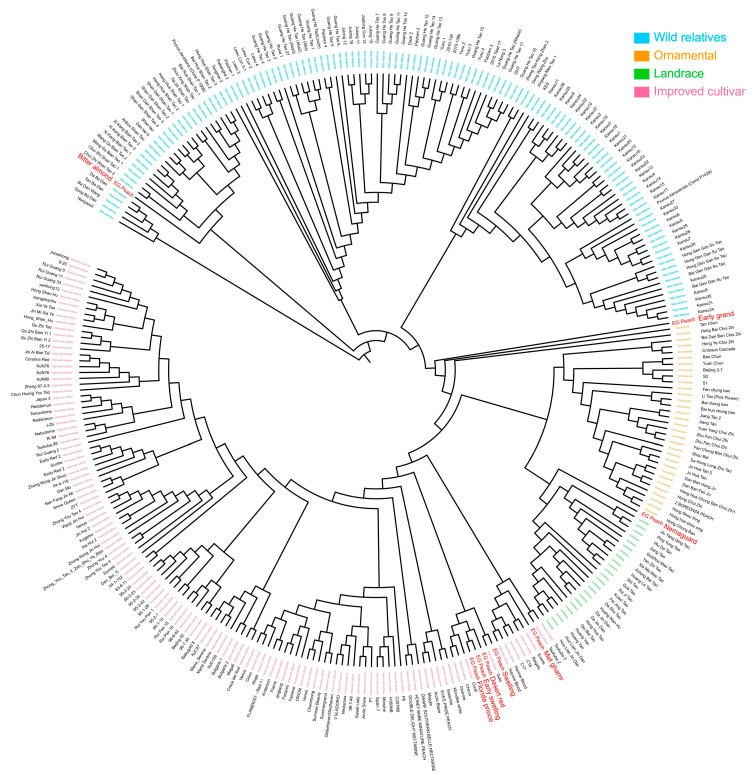
**Phylogenetic analysis of Egyptian peach cultivars and other peach germplasm resources.** The phylogenetic tree illustrates the genetic relationships between 309 peach accessions, including 8 Egyptian cultivars and 301 other germplasm resources, based on a robust dataset of 8,701,042 high-quality SNPs. The accessions are clustered into four distinct groups: wild relatives (cyan), ornamental peach (orange), landraces (green), and improved cultivars (pink). Among them, the red marks in each population represent the peach resources collected in Egypt. The tree was constructed using the MEGA 11 program with 1000 bootstrap replications.

**Figure 6 ijms-25-08497-f006:**
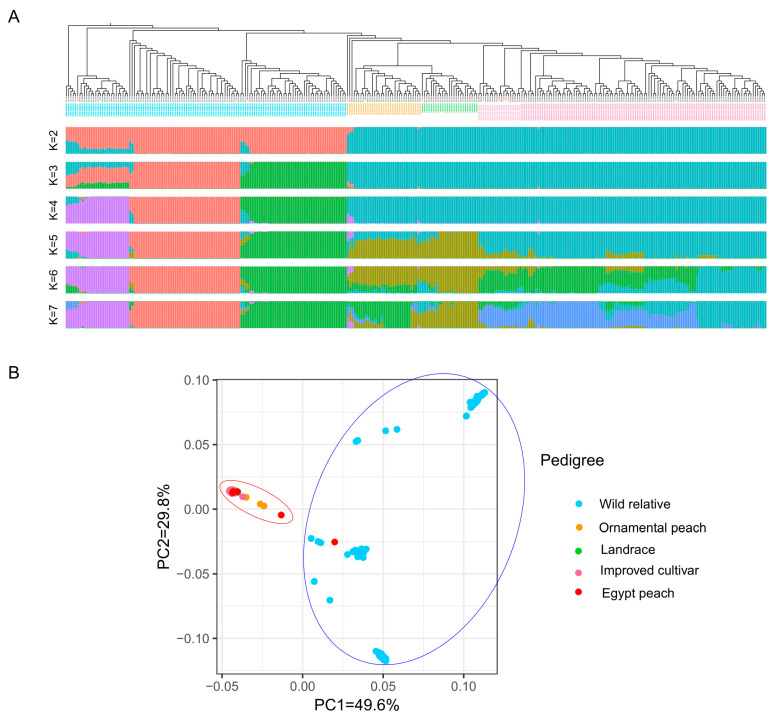
**Population structure and principal component analysis (PCA) of peach genetic diversity**. (**A**) The population structure of 309 peach accessions, including wild relatives (cyan), ornamental peaches (orange), landraces (green), improved cultivars (pink), and Egyptian cultivars (red), was assessed using STRUCTURE analysis with K values ranging from 2 to 7. Each vertical bar represents an individual peach accession, and different colors indicate distinct populations. (**B**) The principal component analysis (PCA) of the 309 peach accessions, based on all identified SNPs, illustrates the genetic diversity within the dataset. Each point represents a single peach accession positioned in a two-dimensional space defined by the first two principal components (PC1 and PC2), which capture a significant portion of the genetic variation.

**Figure 7 ijms-25-08497-f007:**
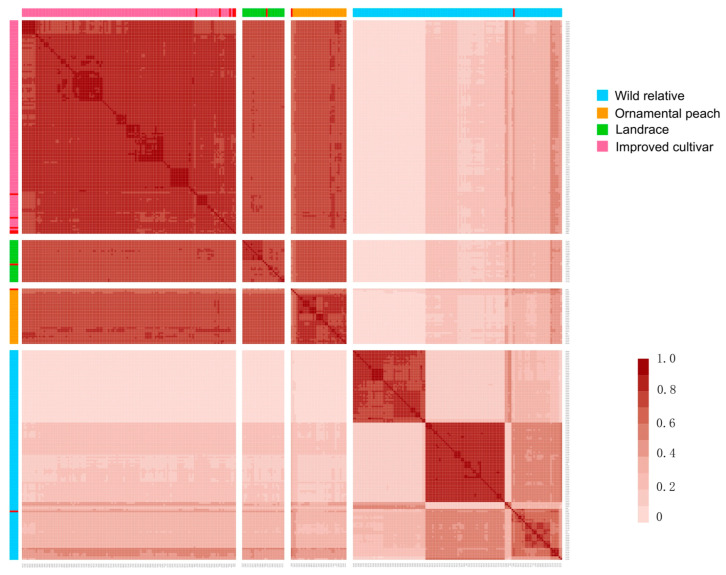
**Relative IBD values between pairwise comparisons in all the tested accessions.** The heatmap showcases the statistical analysis results of IBD between two tested samples. The relative value of IBD ranges from 0 to 1, with 0 indicating a near absence of relationship between the samples. As the value increases, so does the proximity of their connection, resulting in a deeper hue depicted in the above figure. The squares with different colors in the horizontal and vertical coordinates represent distinct groups of peaches, among which 8 Egyptian peaches are highlighted in red. The samples represented by the sample code correspond to those provided in the attached table.

**Table 1 ijms-25-08497-t001:** Sample information of collected Egyptian peach cultivars.

Cultivar	Governorate	Chilling Hours	Ripening Time	Latitude (N)	Longitude (E)
Florida prince	Alexandria	100–150	First week of April	30°33′36.7″	30°15′18.0″
Early swelling	Alexandria	150–250	Third week of April	30°33′33.8″	30°15′32.4″
Early grand	North Sinai	150–250	Late April	31°11′26.1″	34°04′42.5″
Desert red	Alexandria	200–250	Mid-May	30°33′47.7″	30°15′28.8″
Swelling	Alexandria	200–250	First week of June	30°32′53.2″	30°15′16.2″
Met ghamr	Al-Qalyubia	450	Mid-June	30°22′37.4″	31°07′25.0″
Nemaguard	Alexandria	-	-	30°33′47.7″	30°15′28.8″
Bitter almond	North Sinai	-	-	31°12′12.7″	34°02′14.1″

**Table 2 ijms-25-08497-t002:** Summary of the sequencing results of the Egyptian peach cultivars.

Cultivar	Clean Reads	Clean Bases	Q30 (%)	GC (%)	Mapped (%)	Sequencing Coverage (×)	Coverage Rate (1×)
Met ghamr	22,723,292	6,735,864,568	93.4	40.12	93.37	26.07	92.14
Florida prince	22,765,569	6,757,607,956	93.44	39.32	94.61	30.12	98.04
Early swelling	22,359,252	6,650,374,896	93.60	38.53	96.22	29.43	98.23
Early grand	21,192,361	6,296,956,610	92.78	37.61	93.15	29.21	92.88
Desert red	21,002,961	6,249,157,770	96.32	37.65	95.38	28.56	96.97
Swelling	21,488,706	6,380,695,324	92.57	38.33	95.69	28.56	97.12
Nemaguard	19,692,875	5,851,569,914	92.69	39.00	93.08	26.58	93.09
Bitter almond	19,833,337	5,885,734,930	92.40	38.08	92.48	28.44	91.32

## Data Availability

The data presented in this study are available upon request from the corresponding author.

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
