# Peer review of "Origins and Genetic Characteristics of Egyptian Peach"

_ijms, 2024, doi:10.3390/ijms25158497_

Round 1

Reviewer 1 Report

Comments and Suggestions for Authors

1.      The abstract effectively outlines the research objectives related to exploring the origins and genetic characteristics of Egyptian peach. It summarizes the major findings.
2.      The introduction effectively highlights the topic of the manuscript and effectively outlines the scope of the study. However, it could clarify actual knowledge of genetics of Egyptian peaches as well as the specific research objectives guiding the investigation. Clearly stating the primary research questions or hypotheses would give readers a better understanding of the study's purpose and direction.  The introduction is generally well-written, with clear and concise language. However, some sentences could be more dense and benefit from simplification or restructuring for improved readability. 
3.     The description of the study area and sample collection procedures is adequately provided. 
4. Results are written well as well as dicussion, but here again, some simplification of the senteses need to be done. 

Comments on the Quality of English Language

Simplify the syntax.

Author Response

Response to Reviewer 1 Comments

1. Summary

Thank you very much for taking the time to review this manuscript. Please find the detailed responses below and the corresponding revisions in the re-submitted files.

2. Questions for General Evaluation

Reviewer’s Evaluation

Response and Revisions

Does the introduction provide sufficient background and include all relevant references?

    Can be improved

Is the research design appropriate?

          Yes

Are the methods adequately described?

    Can be improved

Are the results clearly presented?

    Can be improved

Are the conclusions supported by the results?

    Can be improved

3. Point-by-point response to Comments and Suggestions for Authors

Comment 1: [The abstract effectively outlines the research objectives related to exploring the origins and genetic characteristics of Egyptian peach. It summarizes the major findings]

Response 1: [Thank you so much for your motivational comment on the abstract of our manuscript. We are glad you found the outline of our research objectives and the summary of our major findings to be clear and effective]

Comment 2: [The introduction effectively highlights the topic of the manuscript and effectively outlines the scope of the study. However, it could clarify actual knowledge of genetics of Egyptian peaches as well as the specific research objectives guiding the investigation. Clearly stating the primary research questions or hypotheses would give readers a better understanding of the study's purpose and direction.  The introduction is generally well-written, with clear and concise language. However, some sentences could be more dense and benefit from simplification or restructuring for improved readability]

Response 2: [Thank you for your constructive feedback on our manuscript's introduction. We appreciate your positive comments and your suggestions for improvement. In response to your feedback, we have clarified the existing knowledge on the genetics of Egyptian peaches, clearly stated the primary research objectives guiding our investigation, and simplified some sentences for improved readability. These changes can be found on page 3, in the last paragraph of the introduction section, and lines 83-95]

Comment 3: [The description of the study area and sample collection procedures is adequately provided]

Response 3: [Thank you so much for your positive feedback. We are pleased to know that you found this section of our manuscript to be adequately provided]

Comment 4: [Results are written well as well as discussion, but here again, some simplification of the sentences need to be done]

Response 4: [Thank you so much for your valuable comment, we appreciate your positive remarks about how the results and discussion sections are written. In response to your comment regarding the simplification of sentences, we have reviewed these sections and refined the language to be clearer and more concise]

4. Response to Comments on the Quality of English Language

Point 1: Simplify the syntax.

Response 1: We appreciate your suggestion to simplify the syntax in our manuscript. Accordingly, we have reviewed the entire text, focusing particularly on areas where the syntax was complex. We have restructured sentences to enhance clarity and readability.

Reviewer 2 Report

Comments and Suggestions for Authors

well written article that thoroughly investigates the diversity of prunus persica and makes interesting hypotheses on the origins of the different varieties. On the basis of the analyzes carried out, the authors find information that suggests a very ancient introduction of fishing in Egypt and this probably allowed a long selection to adapt it to the local climate. I would only suggest delving a little deeper into the history of its diffusion. in this regard, in the summary there is interesting information that is not developed in the subsequent text (e.g. diffusion in Roman times, etc.). finally, some considerations could be made on the fact that by worsening climate change, Egypt could supply and encourage the spread of LCR varieties to other Mediterranean countries. some typos in the attached file.

Author Response

Response to Reviewer 2 Comments

1. Summary

Thank you very much for taking the time to review this manuscript. Please find the detailed responses below and the corresponding revisions in the re-submitted files.

2. Questions for General Evaluation

Reviewer’s Evaluation

Response and Revisions

Does the introduction provide sufficient background and include all relevant references?

          Yes

Is the research design appropriate?

          Yes

Are the methods adequately described?

          Yes

Are the results clearly presented?

          Yes

Are the conclusions supported by the results?

          Yes

3. Point-by-point response to Comments and Suggestions for Authors

Comment 1: [well written article that thoroughly investigates the diversity of prunus persica and makes interesting hypotheses on the origins of the different varieties. On the basis of the analyzes carried out, the authors find information that suggests a very ancient introduction of fishing in Egypt and this probably allowed a long selection to adapt it to the local climate. I would only suggest delving a little deeper into the history of its diffusion. in this regard, in the summary there is interesting information that is not developed in the subsequent text (e.g. diffusion in Roman times, etc.). finally, some considerations could be made on the fact that by worsening climate change, Egypt could supply and encourage the spread of LCR varieties to other Mediterranean countries. some typos in the attached file]

Response 1: [Thank you so much for your motivational comments on our manuscript. We appreciate your valuable suggestions for further improving our paper. In response to your feedback, we have included the history of the diffusion of peach cultivation in Egypt, particularly during Roman times, which can be found on page 8, lines 292-304. We have also added how worsening climate change may position Egypt as a critical supplier of low chilling requirement (LCR) peach varieties to other Mediterranean countries, which can be found on page 8, in the last paragraph of the discussion section, and lines 308-314. Regarding the correction of typos, we have included the author's name for species and have corrected the typos identified in the attached file]
